# A Universal Self-Attention Graph Neural Network

## Abstract

Existing graph embedding models often have weaknesses in exploiting graph structure similarities, potential dependencies among nodes and global network properties. To this end, we present U2GNN, a novel embedding model leveraging on the strength of the recently introduced universal self-attention network (Dehghani et al., 2019), to learn low-dimensional embeddings of graphs which can be used for graph classification. In particular, given an input graph, U2GNN first applies a self-attention computation, which is then followed by a recurrent transition to iteratively memorize its attention on vector representations of each node and its neighbors across each iteration. Thus, U2GNN can address the weaknesses in the existing models in order to produce plausible node embeddings whose sum is the final embedding of the whole graph. Experimental results in both supervised and unsupervised fashions show that our U2GNN produces new state-of-the-art performances on a range of well-known benchmark datasets for the graph classification task.

## 1 Introduction

Many real-world and scientific data are represented in forms of graphs, e.g. data from knowledge graphs, recommender systems, social and citation networks as well as telecommunication and biological networks (Battaglia et al., 2018; Zhang et al., 2018c). In general, a graph can be viewed as a network of nodes and edges, where nodes correspond to individual objects and edges encode relationships among those objects. For example, in online forum, each discussion thread can be constructed as a graph where nodes represent users and edges represent commenting activities between users (Yanardag & Vishwanathan, 2015).

Early approaches focus on computing the similarities among graphs to build a graph kernel for graph classification (Gärtner et al., 2003; Kashima et al., 2003; Borgwardt & Kriegel, 2005; Shervashidze et al., 2009; Vishwanathan et al., 2010; Shervashidze et al., 2011; Yanardag & Vishwanathan, 2015; Narayanan et al., 2017; Ivanov & Burnaev, 2018). These graph kernel-based approaches treat each atomic substructure (e.g., subtree structure, random walk or shortest path) as an individual feature, and count their frequencies to construct a numerical vector to represent the entire graph, hence they ignore *node attributes*, substructure similarities and global network properties.

One recent notable strand is to learn low-dimensional continuous embeddings of the whole graphs (Hamilton et al., 2017b; Zhang et al., 2018a; Zhou et al., 2018), and then use these learned embeddings to train a classifier to predict graph labels (Wu et al., 2019). Advanced approaches in this direction have attempted to exploit graph neural network (Scarselli et al., 2009), capsule network (Sabour et al., 2017) or graph convolutional neural network (Kipf & Welling, 2017; Hamilton et al., 2017a) for supervised learning objectives (Li et al., 2016; Niepert et al., 2016; Zhang et al., 2018b; Ying et al., 2018; Verma & Zhang, 2018; Xu et al., 2019; Xinyi & Chen, 2019; Maron et al., 2019b; Chen et al., 2019). These graph neural network (GNN)-based approaches usually consist of two common phases: the propagating phase and the readout phase. The former phase aims to iteratively update vector representation of each node by recursively aggregating representations of its neighbors, and then the latter phase applies a pooling function (e.g., mean, max or sum pooling) on output node representations to produce an embedding of each entire graph; and this graph embedding is used to predict the graph label. We find that these approaches are currently showing very promising performances, nonetheless the dependency aspect among nodes, which often exhibit strongly

in many kinds of real-world networks, has not been exploited effectively due to *lack of advanced computations within the propagating phase.*

Very recently, the universal self-attention network (Dehghani et al., 2019) has been shown to be very powerful in NLP tasks such as question answering, machine translation and language modeling. Inspired by this new attention network, we propose U2GNN – a novel universal self-attention graph neural network to learn plausible node and graph embeddings. Our intuition comes from an observation that the recurrent attention process in the universal self-attention network can memorize implicit dependencies between each node and its neighbors from previous iterations, which can be then aggregated to further capture the dependencies among substructures into latent representations in subsequent iterations; this process, hence, can capture both local and global graph structures. Algorithmically, at each timestep, our proposed U2GNN iteratively exchanges a node representation with its neighborhood representations using a self-attention mechanism (Vaswani et al., 2017) followed by a recurrent transition to infer node embeddings. We finally take the sum of all learned node embeddings to obtain the embedding of the whole graph. Our main contributions are as follows:

- In our proposed U2GNN, the novelty of memorizing the dependencies among nodes implies that U2GNN can explore the graph structure similarities locally and globally – an important feature that most of existing approaches are unable to do.

- U2GNN can be seen as a general framework where we prove the powerfulness of our model in both supervised or unsupervised fashions. The experimental results on 9 well-known benchmark datasets for the graph classification task show that both our supervised and unsupervised U2GNN models produce new state-of-the-art (SOTA) accuracies in most of benchmark cases.

- To our best of knowledge, our work is the first to show that a unsupervised model can noticeably outperform up-to-date supervised approaches by a large margin. Therefore, we suggest that future GNN works should pay more attention to the unsupervised fashion as well as not comparing supervised models with unsupervised models together. This is important in both industry and academic applications in reality where expanding unsupervised GNN models is more suitable due to the limited availability of class labels.

## 2 RELATED WORK

Early popular approaches are based on "graph kernel" which aims to recursively decompose each graph into "atomic substructures" (e.g., graphlets, subtree structures, random walks or shortest paths) in order to measure the similarity between two graphs (Gärtner et al., 2003). For this reason, we can view each atomic substructure as a word token and each graph as a text document, hence we represent a collection of graphs as a document-term matrix which describes the normalized frequency of terms in documents. Then, we can use a dot product to compute the similarities among graphs to derive a kernel matrix used to measure the classification performance using a kernel-based learning algorithm such as Support Vector Machines (SVM) (Hofmann et al., 2008). We refer to an overview of the graph kernel-based approaches in (Nikolentzos et al., 2019; Kriege et al., 2019).

Since the introduction of word embedding models i.e., Word2Vec (Mikolov et al., 2013) and Doc2Vec (Le & Mikolov, 2014), there have been several efforts attempted to apply them for the graph classification task. Deep Graph Kernel (DGK) (Yanardag & Vishwanathan, 2015) applies Word2Vec to learn embeddings of atomic substructures to create the kernel matrix. Graph2Vec (Narayanan et al., 2017) employs Doc2Vec to obtain embeddings of entire graphs in order to train a SVM classifier to perform classification. Anonymous Walk Embedding (AWE) (Ivanov & Burnaev, 2018) maps random walks into "anonymous walks" which are considered as word tokens, and then utilizes Doc2Vec to achieve the graph embeddings to produce the kernel matrix.

In parallel, another recent line of work has focused on using deep neural networks to perform the graph classification in a supervised manner. PATCHY-SAN (Niepert et al., 2016) adapts a graph labeling procedure to generate a fixed-length sequence of nodes from an input graph, and orders $k$-hop neighbors for each node in the generated sequence according to their graph labelings; PATCHY-SAN then selects a fixed number of ordered neighbors for each node and applies a convolutional neural network to classify the input graph. MPNN (Gilmer et al., 2017), DGCNN (Zhang et al., 2018b) and DIFFPOOL (Ying et al., 2018) are end-to-end supervised models which share similar two-phase process by (i) using stacked multiple graph convolutional layers (e.g., GCN layer (Kipf & Welling,

2017) or GraphSAGE layer (Hamilton et al., 2017a)) to aggregate node feature vectors, and (ii) applying a graph-level pooling layer (e.g., mean, max or sum pooling, sort pooling or differentiable pooling) to obtain the graph embeddings which are then fed to a fully-connected layer followed by a softmax layer to predict the graph labels.

Graph neural networks (GNNs) (Scarselli et al., 2009) aim to iteratively update the vector representation of each node by recursively propagating the representations of its neighbors using a recurrent function until convergence. The recurrent function can be a neural network e.g., gated recurrent unit (GRU) (Li et al., 2016), or multi-layer perceptron (MLP) (Xu et al., 2019). Note that both stacked GCN and GraphSAGE multiple layers can be seen as variants of the recurrent function in GNNs. Other graph embedding models are briefly summarized in (Zhou et al., 2018; Zhang et al., 2018c; Wu et al., 2019).

## 3 THE PROPOSED U2GNN

In this section, we detail how to construct our U2GNN and then present how U2GNN learns model parameters to produce node and graph embeddings.

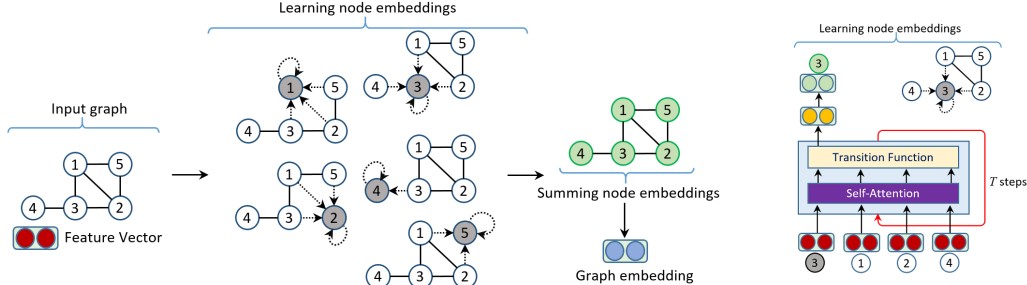

Figure 1: Illustration of our U2GNN learning process, e.g., for node 3 with $d = 2$ and $N = 3$.

**Graph classification.** Given a set of graphs $\{\mathcal{G}_1, \mathcal{G}_2, ..., \mathcal{G}_N\} \subseteq \mathsf{G}$ and their corresponding class labels $\{\mathsf{y}_1, \mathsf{y}_2, ..., \mathsf{y}_N\} \subseteq \mathcal{Y}$, our U2GNN aims to learn a plausible embedding $\mathbf{o}_\mathcal{G}$ of each entire graph $\mathcal{G}$ in order to predict its label $\mathsf{y}$.

Each graph $\mathcal{G}$ is defined as $\mathcal{G} = (\mathcal{V}, \mathcal{E}, \mathbf{X})$, where $\mathcal{V}$ is a set of nodes, $\mathcal{E}$ is a set of edges, and $\mathbf{X} \in \mathbb{R}^{|\mathcal{V}| \times d}$ represents feature vectors of nodes. In U2GNN, as illustrated in Figure 1, we use a universal self-attention network (Dehghani et al., 2019) to learn a node embedding $\mathbf{o}_\mathsf{v}$ of each node $\mathsf{v} \in \mathcal{V}$, and then $\mathbf{o}_\mathcal{G}$ is simply returned by summing all learned node embeddings as follows:[1]

$$\mathbf{o}_\mathcal{G} = \text{SUM}\left(\{\mathbf{o}_\mathsf{v}, \forall \mathsf{v} \in \mathcal{V}\}\right) \tag{1}$$

**Constructing U2GNN.** Formally, given an input graph $\mathcal{G}$, we uniformly sample a set of $N$ neighbors for each $\mathsf{v} \in \mathcal{V}$, and then use node $\mathsf{v}$ and its neighbors for the U2GNN learning process.[2] For example, as illustrated in Figure 1, we generate a set of $N = 3$ neighbors $\{1, 2, 4\}$ for node 3, and then consider $\{3, 1, 2, 4\}$ as an input to U2GNN where we leverage on the universal self-attention network (Dehghani et al., 2019) to learn an effective embedding of node 3.

Intuitively, the universal self-attention network can help to better aggregate feature vectors from neighbors of a given node to produce its plausible embedding. In particular, each node and its neighbors is transformed into a sequence of feature vectors which are then iteratively refined at each timestep – using a self-attention mechanism (Vaswani et al., 2017) followed by a recurrent transition (TRANS) with adding residual connection (He et al., 2016) and layer normalization (LNORM) (Ba et al., 2016).

Given a node $\mathsf{v} \in \mathcal{V}$, we consider a sequence of $(N + 1)$ nodes $\{\mathsf{v}_i\}_{i=1}^{N+1}$ where $\mathsf{v}_1 = \mathsf{v}$ and $\{\mathsf{v}_i\}_{i=2}^{N+1}$ are $N$ sampled neighbors of $\mathsf{v}$. We obtain an input sequence of feature vectors $\{\mathbf{h}_{\mathsf{v}_i}^0\}_{i=1}^{N+1}$ for which

---

[1]The experimental results in (Xu et al., 2019) show that the sum pooling performs better than the mean and max poolings.

[2]We sample a different set of neighbors at each training step.

$\mathbf{h}_{v_i}^0 = \mathbf{X}_{v_i} \in \mathbb{R}^d$. In U2GNN, at each step $t$, we feed $\{\mathbf{h}_{v_i}^{t-1}\}_{i=1}^{N+1}$ as an input sequence and produce an output sequence $\{\mathbf{h}_{v_i}^t\}_{i=1}^{N+1}$ for which $\mathbf{h}_{v_i}^t \in \mathbb{R}^d$ as follows:

$$\mathbf{h}_{v_i}^t = \text{LNORM}\left(\mathbf{x}_{v_i}^t + \text{TRANS}\left(\mathbf{x}_{v_i}^t\right)\right) \tag{2}$$

$$\text{with } \mathbf{x}_{v_i}^t = \text{LNORM}\left(\mathbf{h}_{v_i}^{t-1} + \text{ATT}\left(\mathbf{h}_{v_i}^{t-1}\right)\right) \tag{3}$$

where $\text{TRANS}(.)$ and $\text{ATT}(.)$ denote a feed-forward network and a self-attention network respectively as follows:

$$\text{TRANS}\left(\mathbf{x}_{v_i}^t\right) = \mathbf{W}_2 \text{ReLU}\left(\mathbf{W}_1 \mathbf{x}_{v_i}^t + \mathbf{b}_1\right) + \mathbf{b}_2 \tag{4}$$

where $\mathbf{W}_1 \in \mathbb{R}^{k \times d}$ and $\mathbf{W}_2 \in \mathbb{R}^{d \times k}$ are weight matrices, and $\mathbf{b}_1$ and $\mathbf{b}_2$ are bias parameters, and:

$$\text{ATT}\left(\mathbf{h}_{v_i}^{t-1}\right) = \sum_{j=1}^{N+1} \alpha_{i,j}\left(\boldsymbol{V}\mathbf{h}_{v_j}^{t-1}\right) \tag{5}$$

where $\boldsymbol{V} \in \mathbb{R}^{d \times d}$ is a value-projection weight matrix; $\alpha_{i,j}$ is an attention weight which is computed using the softmax function over scaled dot products between the $i$-th and $j$-th input nodes:

$$\alpha_{i,j} = \text{softmax}\left(\frac{\left(\boldsymbol{Q}\mathbf{h}_{v_i}^{t-1}\right) \cdot \left(\boldsymbol{K}\mathbf{h}_{v_j}^{t-1}\right)}{\sqrt{d}}\right) \tag{6}$$

where $\boldsymbol{Q} \in \mathbb{R}^{d \times d}$ and $\boldsymbol{K} \in \mathbb{R}^{d \times d}$ are query-projection and key-projection matrices, respectively.[3]

---

**Algorithm 1:** The unsupervised learning process.

1 **Input**: $\mathcal{G} = (\mathcal{V}, \mathcal{E}, \mathbf{X})$
2 **for** $v \in \mathcal{V}$ **do**
3    SAMPLE $\{v_i\}_{i=1}^{N+1}$ where $\{v_i\}_{i=2}^{N+1}$ are $N$ sampled neighbors of $v_1 = v$
4    **for** $t = 1, 2, ..., T$ **do**
5       $\forall i \in \{1, 2, ..., (N+1)\}$
6       $\mathbf{x}_{v_i}^t \leftarrow \text{LNORM}\left(\mathbf{h}_{v_i}^{t-1} + \text{ATT}\left(\mathbf{h}_{v_i}^{t-1}\right)\right)$
7       $\mathbf{h}_{v_i}^t \leftarrow \text{LNORM}\left(\mathbf{x}_{v_i}^t + \text{TRANS}\left(\mathbf{x}_{v_i}^t\right)\right)$
8 $\mathbf{o}_v \leftarrow \mathbf{h}_v^T, \forall v \in \mathcal{V}$
9 $\mathbf{o}_{\mathcal{G}} = \text{SUM}\left(\{\mathbf{o}_v, \forall v \in \mathcal{V}\}\right)$

**Algorithm 2:** The supervised learning process.

1 **Input**: $\mathcal{G} = (\mathcal{V}, \mathcal{E}, \mathbf{X})$ with its label y
2 **for** $v \in \mathcal{V}$ **do**
3    SAMPLE $\{v_i\}_{i=1}^{N+1}$ where $\{v_i\}_{i=2}^{N+1}$ are $N$ sampled neighbors of $v_1 = v$
4    **for** $t = 1, 2, ..., T$ **do**
5       $\forall i \in \{1, 2, ..., (N+1)\}$
6       $\mathbf{x}_{v_i}^t \leftarrow \text{LNORM}\left(\mathbf{h}_{v_i}^{t-1} + \text{ATT}\left(\mathbf{h}_{v_i}^{t-1}\right)\right)$
7       $\mathbf{h}_{v_i}^t \leftarrow \text{LNORM}\left(\mathbf{x}_{v_i}^t + \text{TRANS}\left(\mathbf{x}_{v_i}^t\right)\right)$
8 $\mathbf{o}_{\mathcal{G}} = \text{SUM}\left(\{\mathbf{h}_v^T, \forall v \in \mathcal{V}\}\right)$
9 y $\leftarrow \mathbf{W}\mathbf{o}_{\mathcal{G}} + \mathbf{b}$

---

**Learning parameters of U2GNN in "unsupervised" fashion:** After $T$ steps, we use the vector representation $\mathbf{h}_{v_1}^T$ to infer node embeddings $\mathbf{o}_{v_1}$. For example, as shown in Figure 1, we have $v_1 = 3$, $v_2 = 1$, $v_3 = 2$ and $v_4 = 4$, and then consider $\mathbf{h}_3^T$ to infer $\mathbf{o}_3$. We learn our model parameters (including the weight matrices and biases as well as node embeddings $\mathbf{o}_v$) by minimizing the sampled softmax loss function (Jean et al., 2015) applied to node $v \in \mathcal{V}$ as follows:

$$\mathcal{L}_{\text{U2GNN}}(v) = -\log \frac{\exp(\mathbf{o}_v \cdot \mathbf{h}_v^T)}{\sum_{v' \in \mathcal{V}'} \exp(\mathbf{o}_{v'} \cdot \mathbf{h}_v^T)} \tag{7}$$

where $\mathcal{V}'$ is a subset sampled from $\mathcal{V}$.

We briefly describe the general learning process of our proposed U2GNN model in Algorithm 1. Here, the learned node embeddings $\mathbf{o}_v$ are used as the final representations of nodes $v \in \mathcal{V}$. After that, we obtain the plausible embedding $\mathbf{o}_{\mathcal{G}}$ of the graph $\mathcal{G}$ by summing all learned node embeddings as mentioned in Equation 1. Finally, we employ a logistic regression classifier from LIBLINEAR (Fan et al., 2008) to predict the graph labels.

---

[3]It is important to note that GAT (Veličković et al., 2018) borrows the standard attention technique from (Bahdanau et al., 2015) in using a single-layer feedforward neural network parametrized by a weight vector and then applying the LeakyReLU non-linearity function followed by the softmax function to compute importance weights of neighbors of a given node. Therefore, *GAT is much different with the self-attention mechanism.* More information about GCN, GraphSAGE and GAT can be found in Appendix A.

**Learning parameters of U2GNN in "supervised" fashion:** We do not need to learn the node embeddings separately, hence after $T$ steps, we use $\mathbf{h}_{\mathsf{v}_1}^T$ as the final embedding of node $\mathsf{v}_1 \in \mathcal{V}$ (i.e., $\mathbf{o}_\mathsf{v} = \mathbf{h}_\mathsf{v}^T$). We then feed $\mathbf{o}_\mathcal{G}$ to a fully connected layer to predict the graph labels as briefly presented in Algorithm 2. Finally, we learn the model parameters by minimizing the cross entropy loss function.

**Proof**: In general, on the node level, each node and its neighbors are iteratively attended in the recurrent process with weight matrices shared across timesteps and iterations, thus U2GNN can memorize the potential dependencies among nodes within substructures. On the graph level, U2GNN views the shared weight matrices as memories to access the updated node-level information from previous iterations to further aggregate broader dependencies among substructures into implicit graph representations in subsequent iterations. Therefore, U2GNN is advantageous to capture both global and local graph structures to learn effective node and graph embeddings, leading to state-of-the-art performances for the graph classification task.

## 4 EXPERIMENTAL SETUP

We prove the effectiveness of our U2GNN on the graph classification task using a range of well-known benchmark datasets for both supervised and unsupervised fashions.

### 4.1 DATASETS

We use 9 well-known datasets consisting of 5 social network datasets (COLLAB, IMDB-B, IMDB-M, RDT-B and RDT-M5K) (Yanardag & Vishwanathan, 2015) and 4 bioinformatics datasets (DD, MUTAG, PROTEINS and PTC). We follow (Niepert et al., 2016; Zhang et al., 2018b) to use node degrees as features on all social network datasets as these datasets do not have available node features. Table 1 reports the statistics of these datasets.

| Dataset | #G | #Cls | A.NG | A.NN | $d$ |
|---|---|---|---|---|---|
| COLLAB | 5,000 | 3 | 74.5 | 65.9 | – |
| IMDB-B | 1,000 | 2 | 19.8 | 9.8 | – |
| IMDB-M | 1,500 | 3 | 13.0 | 10.1 | – |
| RDT-B | 2,000 | 2 | 429.6 | 2.3 | – |
| RDT-M5K | 5,000 | 5 | 508.5 | 2.3 | – |
| DD | 1,178 | 2 | 284.3 | 5.0 | 82 |
| MUTAG | 188 | 2 | 17.9 | 2.2 | 7 |
| PROTEINS | 1,113 | 2 | 39.1 | 3.7 | 3 |
| PTC | 344 | 2 | 25.6 | 2.0 | 19 |

Table 1: Statistics of the experimental benchmark datasets. **#G** denotes the numbers of graphs. **#Cls** denotes the number of graph classes. **A.NG** denotes the average number of nodes per graph. **A.NN** denotes the average number of neighbors per node. $d$ is the dimension of node feature vectors (i.e. the number of node labels).

**Social networks datasets.** COLLAB is a scientific dataset where each graph represents a collaboration network of a corresponding researcher with other researchers from each of 3 physics fields; and each graph is labeled to a physics field the researcher belong to. IMDB-B and IMDB-M are movie collaboration datasets where each graph is derived from actor/actress and genre information of different movies on IMDB, in which nodes correspond to actors/actresses, and each edge represents a co-appearance of two actors/actresses in the same movie; and each graph is assigned to a genre. RDT-B and RDT-M5K are datasets derived from Reddit community, in which each online discussion thread is viewed as a graph where nodes correspond to users, two users are linked if at least one of them replied to another's comment; and each graph is labeled to a sub-community the corresponding thread belongs to.

**Bioinformatics datasets.** DD (Dobson & Doig, 2003) is a collection of 1,178 protein network structures with 82 discrete node labels, where each graph is classified into enzyme or non-enzyme class. PROTEINS comprises 1,113 graphs obtained from (Borgwardt et al., 2005) to present secondary structure elements (SSEs). MUTAG (Debnath et al., 1991) is a collection of 188 nitro compound networks with 7 discrete node labels, where classes indicate a mutagenic effect on a bacterium. PTC (Toivonen et al., 2003) consists of 344 chemical compound networks with 19 discrete node labels where classes show carcinogenicity for male and female rats.

### 4.2 TRAINING PROTOCOL TO LEARN GRAPH EMBEDDINGS

**Coordinate embedding:** The relative coordination among nodes might provide meaningful information about graph structure. We follow Dehghani et al. (2019) to associate each position $i$ at step $t$ a pre-defined coordinate embedding $\mathbf{p}_i^t$ using the sinusoidal functions (Vaswani et al., 2017), thus we can change Equation 3 in Section 3 to:

$$
\begin{aligned}
\mathbf{x}_{\mathsf{v}_i}^t &= \text{LNORM}\left(\mathbf{h}_{\mathsf{v}_i}^{t-1} + \text{ATT}\left(\mathbf{h}_{\mathsf{v}_i}^{t-1} + \mathbf{p}_i^t\right)\right) \quad (8) \\
\text{with } \mathsf{p}_{i,2j}^t &= \sin(i/10000^{2j/d}) + \sin(t/10000^{2j/d}) \\
\mathsf{p}_{i,2j+1}^t &= \cos(i/10000^{2j/d}) + \cos(t/10000^{2j/d})
\end{aligned}
$$

From the preliminary experiments, adding coordinate embeddings enhances classification results on MUTAG and PROTEINS in the unsupervised fashion, hence we use the coordinate embeddings only for these two datasets.

**Hyper-parameter setting to learn our model parameters for all experimental datasets:** (i) Regarding the unsupervised fashion, we fix the hidden size of the feed-forward network in Equation 4 to 1024 ($k = 1024$), and the number of samples in the sampled loss function $\mathcal{L}_{\text{U2GNN}}$ to 512 ($|\mathcal{V}'| = 512$) in Equation 7. We set the batch size to 512 for COLLAB, DD and RDT-B; 1024 for RDT-M5K; and 128 for remaining datasets. (ii) Regarding the supervised fashion, we vary the the hidden size $k$ in $\{32, 128, 512, 1024\}$ and the batch size in $\{1, 2, 4\}$. (iii) For both the unsupervised and supervised fashions, we use the number $N$ of neighbors sampled for each node from $\{4, 8, 16\}$ and the number $T$ of steps from $\{1, 2, 3, 4, 5, 6\}$. We apply the Adam optimizer (Kingma & Ba, 2014) to train our U2GNN model and apply a grid search to select the Adam initial learning rate $lr \in \{5e^{-5}, 1e^{-4}, 5e^{-4}, 1e^{-3}\}$. We run up to 50 epochs and evaluate the model as in what follows.

### 4.3 EVALUATION PROTOCOL

For each dataset, after obtaining the graph embeddings, we perform the same evaluation process from (Yanardag & Vishwanathan, 2015; Niepert et al., 2016; Zhang et al., 2018b; Xu et al., 2019; Xinyi & Chen, 2019), which is using 10-fold cross-validation scheme to calculate the classification performance for a fair comparison.[4]

**Baseline models:** We compare our U2GNN with up-to-date strong baselines as follows:

- **Unsupervised approaches:** Graphlet Kernel (GK) (Shervashidze et al., 2009), Weisfeiler-Lehman kernel (WL) (Shervashidze et al., 2011), Deep Graph Kernel (DGK) (Yanardag & Vishwanathan, 2015) and Anonymous Walk Embedding (AWE) (Ivanov & Burnaev, 2018).

- **Supervised approaches:** PATCHY-SAN (PSCN) (Niepert et al., 2016), Graph Convolutional Network (GCN) (Kipf & Welling, 2017), GraphSAGE (Hamilton et al., 2017a), Deep Graph CNN (DGCNN) (Zhang et al., 2018b), Graph Capsule Convolution Neural Network (GCAPS) (Verma & Zhang, 2018), Capsule Graph Neural Network (CapsGNN) (Xinyi & Chen, 2019), Graph Isomorphism Network (GIN) (Xu et al., 2019), Graph Feature Network (GFN) (Chen et al., 2019), Invariant-Equivariant Graph Network (IEGN) (Maron et al., 2019b), Provably Powerful Graph Network (PPGN) (Maron et al., 2019a) and Discriminative Structural Graph Classification (DSGC) (Seo et al., 2019).

We report the baseline results taken from the original papers or published in (Ivanov & Burnaev, 2018; Verma & Zhang, 2018; Xinyi & Chen, 2019; Chen et al., 2019; Seo et al., 2019).

## 5 EXPERIMENTAL RESULTS

Table 2 presents the experimental results on the 9 benchmark datasets. We use "**Unsupervised**" to denote the unsupervised graph embedding models that can access all nodes from the whole dataset, but not the graph labels; and we use "**Supervised**" to denote the supervised models that use the graph labels of training graphs during training.

---

[4]Regarding the unsupervised fashion, we use the logistic regression classifier from LIBLINEAR (Fan et al., 2008) with setting the termination criterion to 0.001.

In general, our unsupervised U2GNN achieves state-of-the-art performances on a range of benchmarks for the graph classification task, hence this demonstrates a high impact of the unsupervised U2GNN in inferring the plausible node and graph embeddings. It is to note that several existing works (Xinyi & Chen, 2019; Xu et al., 2019; Chen et al., 2019; Maron et al., 2019b; Seo et al., 2019) compared the supervised models with the unsupervised models together due to the use of the same 10-fold cross-validation scheme. *However, our unsupervised U2GNN significantly outperforms the supervised methods in most of benchmark cases by a large margin of 18+%, thus we suggest that future GNN works should separate two training fashions.*

| | Model | COLLAB | IMDB-B | IMDB-M | RDT-B | RDT-M5K |
|---|---|---|---|---|---|---|
| **Unsupervised** | GK (2009) | $72.84 \pm 0.28$ | $65.87 \pm 0.98$ | $43.89 \pm 0.38$ | $77.34 \pm 0.18$ | $41.01 \pm 0.17$ |
| | WL (2011) | $79.02 \pm 1.77$ | $73.40 \pm 4.63$ | $49.33 \pm 4.75$ | $81.10 \pm 1.90$ | $49.44 \pm 2.36$ |
| | DGK (2015) | $73.09 \pm 0.25$ | $66.96 \pm 0.56$ | $44.55 \pm 0.52$ | $78.04 \pm 0.39$ | $41.27 \pm 0.18$ |
| | AWE (2018) | $73.93 \pm 1.94$ | $74.45 \pm 5.83$ | $51.54 \pm 3.61$ | $\textbf{87.89} \pm \textbf{2.53}$ | $50.46 \pm 1.91$ |
| | **U2GNN** | $\textbf{95.62} \pm \textbf{0.92}$ | $\textbf{93.50} \pm \textbf{2.27}$ | $\textbf{74.80} \pm \textbf{4.11}$ | $84.80 \pm 1.53$ | $\textbf{77.25} \pm \textbf{1.46}$ |
| **Supervised** | DSGC (2019) | $79.20 \pm 1.60$ | $73.20 \pm 4.90$ | $48.50 \pm 4.80$ | $92.20 \pm 2.40$ | $-$ |
| | GFN (2019) | $81.50 \pm 2.42$ | $73.00 \pm 4.35$ | $51.80 \pm 5.16$ | $-$ | $\textbf{57.59} \pm \textbf{2.40}$ |
| | PPGN (2019a) | $81.38 \pm 1.42$ | $73.00 \pm 5.77$ | $50.46 \pm 3.59$ | $-$ | $-$ |
| | GIN (2019) | $80.20 \pm 1.90$ | $75.10 \pm 5.10$ | $52.30 \pm 2.80$ | $92.40 \pm 2.50$ | $57.50 \pm 1.50$ |
| | IEGN (2019b) | $77.92 \pm 1.70$ | $71.27 \pm 4.50$ | $48.55 \pm 3.90$ | $-$ | $-$ |
| | CapsGNN (2019) | $79.62 \pm 0.91$ | $73.10 \pm 4.83$ | $50.27 \pm 2.65$ | $-$ | $50.46 \pm 1.91$ |
| | GCAPS (2018) | $77.71 \pm 2.51$ | $71.69 \pm 3.40$ | $48.50 \pm 4.10$ | $87.61 \pm 2.51$ | $50.10 \pm 1.72$ |
| | DGCNN (2018b) | $73.76 \pm 0.49$ | $70.03 \pm 0.86$ | $47.83 \pm 0.85$ | $76.02 \pm 1.73$ | $48.70 \pm 4.54$ |
| | GCN (2017) | $\textbf{81.72} \pm \textbf{1.64}$ | $73.30 \pm 5.29$ | $51.20 \pm 5.13$ | $-$ | $56.81 \pm 2.37$ |
| | GraphSAGE (2017a) | $79.70 \pm 1.70$ | $72.40 \pm 3.60$ | $49.90 \pm 5.00$ | $89.10 \pm 1.90$ | $-$ |
| | PSCN (2016) | $72.60 \pm 2.15$ | $71.00 \pm 2.29$ | $45.23 \pm 2.84$ | $86.30 \pm 1.58$ | $49.10 \pm 0.70$ |
| | **U2GNN** | $77.84 \pm 1.48$ | $\textbf{79.40} \pm \textbf{4.35}$ | $\textbf{56.20} \pm \textbf{3.35}$ | $80.25 \pm 2.38$ | $50.90 \pm 2.31$ |

| | Model | DD | PROTEINS | MUTAG | PTC |
|---|---|---|---|---|
| **Unsupervised** | GK (2009) | $78.45 \pm 0.26$ | $71.67 \pm 0.55$ | $81.58 \pm 2.11$ | $57.26 \pm 1.41$ |
| | WL (2011) | $79.78 \pm 0.36$ | $74.68 \pm 0.49$ | $82.05 \pm 0.36$ | $57.97 \pm 0.49$ |
| | DGK (2015) | $73.50 \pm 1.01$ | $75.68 \pm 0.54$ | $87.44 \pm 2.72$ | $60.08 \pm 2.55$ |
| | AWE (2018) | $71.51 \pm 4.02$ | $-$ | $\textbf{87.87} \pm \textbf{9.76}$ | $-$ |
| | **U2GNN** | $\textbf{95.67} \pm \textbf{1.89}$ | $\textbf{78.07} \pm \textbf{3.36}$ | $81.34 \pm 6.56$ | $\textbf{84.59} \pm \textbf{5.12}$ |
| **Supervised** | DSGC (2019) | $77.40 \pm 6.40$ | $74.20 \pm 3.80$ | $86.70 \pm 7.60$ | $-$ |
| | GFN (2019) | $78.78 \pm 3.49$ | $76.46 \pm 4.06$ | $90.84 \pm 7.22$ | $-$ |
| | PPGN (2019a) | $-$ | $77.20 \pm 4.73$ | $90.55 \pm 8.70$ | $66.17 \pm 6.54$ |
| | GIN (2019) | $-$ | $76.20 \pm 2.80$ | $89.40 \pm 5.60$ | $64.60 \pm 7.00$ |
| | IEGN (2019b) | $-$ | $75.19 \pm 4.30$ | $84.61 \pm 10.0$ | $59.47 \pm 7.30$ |
| | CapsGNN (2019) | $75.38 \pm 4.17$ | $76.28 \pm 3.63$ | $86.67 \pm 6.88$ | $-$ |
| | GCAPS (2018) | $77.62 \pm 4.99$ | $76.40 \pm 4.17$ | $-$ | $66.01 \pm 5.91$ |
| | DGCNN (2018b) | $79.37 \pm 0.94$ | $75.54 \pm 0.94$ | $85.83 \pm 1.66$ | $58.59 \pm 2.47$ |
| | GCN (2017) | $79.12 \pm 3.07$ | $75.65 \pm 3.24$ | $87.20 \pm 5.11$ | $-$ |
| | GraphSAGE (2017a) | $65.80 \pm 4.90$ | $65.90 \pm 2.70$ | $79.80 \pm 13.90$ | $-$ |
| | PSCN (2016) | $77.12 \pm 2.41$ | $75.89 \pm 2.76$ | $\textbf{92.63} \pm \textbf{4.21}$ | $62.29 \pm 5.68$ |
| | **U2GNN** | $\textbf{81.24} \pm \textbf{1.84}$ | $\textbf{78.53} \pm \textbf{4.07}$ | $89.97 \pm 3.65$ | $\textbf{79.36} \pm \textbf{4.06}$ |

Table 2: Graph classification results (% accuracy). The best scores are in **bold**.

Regarding our supervised U2GNN, on the social network datasets, our model produces new state-of-the-art performances on IMDB-B and IMDB-M, especially U2GNN significantly achieves 4+% absolute higher accuracies than all supervised baselines. In addition, U2GNN obtains promising scores on COLLAB, RDT-B and RDT-M5K. On the bioinformatics datasets, our U2GNN obtains new highest accuracies on DD, PROTEINS and PTC. Moreover, U2GNN produces a competitive accuracy compared with those of baseline models on MUTAG, for which there are no significant differences between our supervised U2GNN and some supervised baselines (e.g., GFN, GIN and

PPGN). Note that there are only 188 graphs in the MUTAG dataset, which explains the high variance in the results.

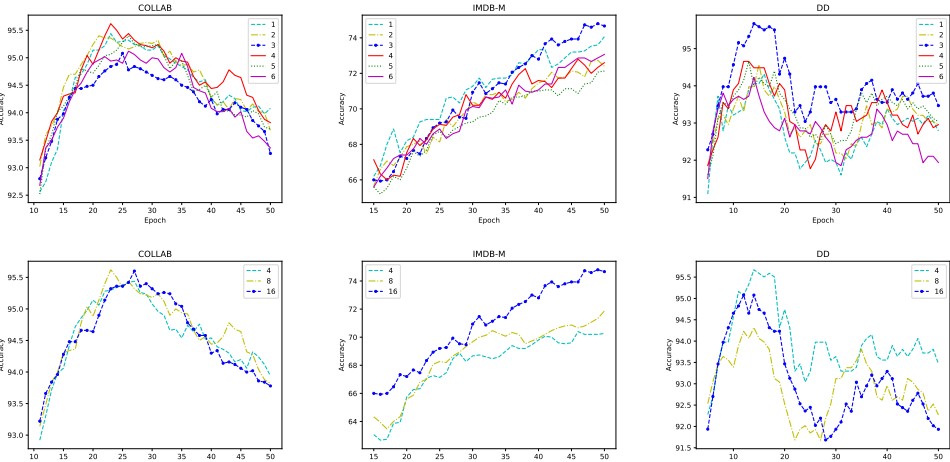

Figure 2: Effects of the number of steps ($T$) (in the 3 top figures), and the number of neighbors ($N$) sampled for each node (in the 3 bottom figures) in the unsupervised fashion. Regarding each dataset, for all 10 folds, we vary the value of either $T$ or $N$ while using the same fixed values of other hyper-parameters.

Next we investigate the effects of hyper-parameters on the experimental datasets in the unsupervised fashion in Figure 2.[5] In general, our unsupervised U2GNN could consistently obtain better results than those of baselines with any value of $T$ and $N$, as long as the training process is stopped precisely for all datasets. In particular, we find that higher $T$ helps on most of the datasets, and especially boosts the performance on bioinformatics data. A possible reason is that the bioinformatics datasets comprise sparse networks where the average number of neighbors per node is below 5 as shown in Table 1, hence we need to use more steps to learn graph properties. In addition, using small $N$ generally produces higher performances on the bioinformatics datasets, while using higher values of $N$ is more suitable for the social network datasets. Besides, the social network datasets are much denser than the bioinformatics datasets, thus this is reason why we should use more sampled neighbors on the social networks rather than the bioinformatics ones. Note that the similar findings also occur in the supervised fashion.

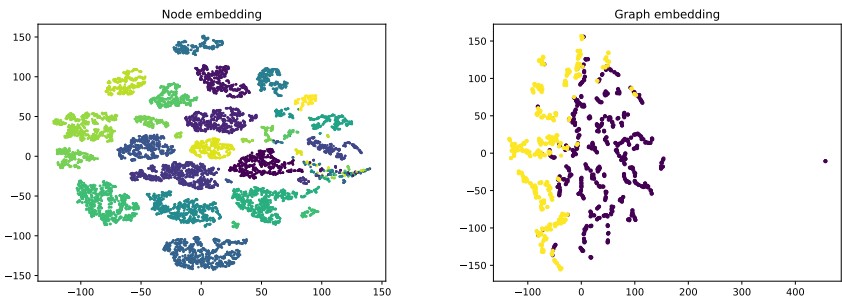

Figure 3: A visualization of the node and graph embeddings learned by our unsupervised U2GNN on the DD dataset. We do not include the embedding visualization of other baselines because those methods are significantly different from our unsupervised U2GNN.

To qualitatively demonstrate the effectiveness of capturing the local and global graph properties, we use t-SNE (Maaten & Hinton, 2008) to visualize the learned node and graph embeddings in the

---

[5]More figures can be found in Appendix B.

unsupervised fashion on the DD dataset where the node labels are available. It can be seen from Figure 3 that our unsupervised U2GNN can effectively capture the local structure wherein the nodes are clustered according to the node labels, and the global structure wherein the graph embeddings are well-separated from each other; verifying the plausibility of the learned node and graph embeddings.

## 6 CONCLUSION

In this paper, we introduce a novel graph neural network model U2GNN for the graph classification task. Given an input graph, U2GNN applies a self-attention mechanism followed by a recurrent transition to learn the node embeddings and then sums all the learned node embeddings to obtain the embedding of the entire graph. Our U2GNN achieves new highest accuracies on most of 9 well-known benchmark datasets in both the supervised and unsupervised fashions, using the same 10-fold cross-validation scheme, against the up-to-date unsupervised and supervised baselines. In future works, we plan to investigate the effectiveness of U2GNN on other important tasks such as node classification and link prediction. Our code is available at: https://anonymous-url/.

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

## A  GNN VARIANTS

Regarding GCN (Kipf & Welling, 2017), each node $v \in \mathcal{V}$ has a feature vector $\boldsymbol{x}_v \in \mathbb{R}^{d_1}$, and GCN infers a new vector representation $\boldsymbol{h}_v \in \mathbb{R}^{d_2}$ for node $v$ from its neighbors as follows:

$$\boldsymbol{h}_v = \mathrm{g}\left(\sum_{u \in \mathcal{N}_v \cup \{v\}} (\boldsymbol{W}\boldsymbol{x}_u + \boldsymbol{b})\right), \forall v \in \mathcal{V} \tag{9}$$

where $\boldsymbol{W} \in \mathbb{R}^{d_2 \times d_1}$ and $\boldsymbol{b} \in \mathbb{R}^{d_2}$ are a weight matrix and a bias respectively, g is a non-linear activation function, and $\mathcal{N}_v$ is the set of neighbors of node $v$. GCN can indirectly capture nodes at many hops away by using multiple GCN layers stacked on top of each other as follows:

$$\boldsymbol{h}_v^{(k+1)} = \mathrm{g}\left(\sum_{u \in \mathcal{N}_v \cup \{v\}} \left(\boldsymbol{W}^{(k)}\boldsymbol{h}_u^{(k)} + \boldsymbol{b}^{(k)}\right)\right), \forall v \in \mathcal{V} \tag{10}$$

where $k$ is the layer index, and $\boldsymbol{h}_{\mathsf{u}}^{(0)} = \boldsymbol{x}_{\mathsf{u}}$. The vector outputs of the last GCN layer are used as the final vector representations of nodes. GCN can be jointly trained with a classification classifier by minimizing the cross-entropy loss function.

GraphSAGE (Hamilton et al., 2017a) is an extension of GCN, in which the representation $\boldsymbol{h}_{\mathsf{v}}^{(k+1)}$ of node v after $k$-th GraphSAGE layer is produced as follows:

$$\boldsymbol{h}_{\mathsf{v}}^{(k+1)} = \mathsf{g}\left(\boldsymbol{W}_{SAGE}^{(k)}\left[\boldsymbol{h}_{\mathsf{v}}^{(k)} \oplus \boldsymbol{h}_{\mathcal{N}_{\mathsf{v}}}^{(k+1)}\right]\right), \forall \mathsf{v} \in \mathcal{V} \tag{11}$$

where $[\oplus]$ denotes a vector concatenation operation, and $\boldsymbol{h}_{\mathcal{N}_{\mathsf{v}}}^{(k+1)}$ can be computed in several ways such as using the GCN computation as follows:

$$\boldsymbol{h}_{\mathcal{N}_{\mathsf{v}}}^{(k+1)} = \mathsf{g}\left(\sum_{\mathsf{u} \in \mathcal{N}_{\mathsf{v}}}\left(\boldsymbol{W}_{GCN}^{(k)}\boldsymbol{h}_{\mathsf{u}}^{(k)} + \boldsymbol{b}^{(k)}\right)\right), \forall \mathsf{v} \in \mathcal{V} \tag{12}$$

or applying an element-wise max-pooling operation as follows:

$$\boldsymbol{h}_{\mathcal{N}_{\mathsf{v}}}^{(k+1)} = \mathsf{max}\left(\{\mathsf{g}\left(\boldsymbol{W}_{pool}\boldsymbol{h}_{\mathsf{u}}^{(k)} + \boldsymbol{b}\right), \forall \mathsf{u} \in \mathcal{N}_{\mathsf{v}}\}\right) \tag{13}$$

In addition, $\boldsymbol{h}_{\mathcal{N}_{\mathsf{v}}}^{(k+1)}$ can be computed by taking the mean (or sum) of all vectors in $\{\boldsymbol{h}_{\mathsf{u}}^{(k)}, \forall \mathsf{u} \in \mathcal{N}_{\mathsf{v}}\}$. $\mathcal{N}_{\mathsf{v}}$ is defined as a fixed-size, uniformly drawn from the set of all neighbor nodes of v, and uniformly sampled differently through each layer. The vector outputs of the last GraphSAGE layer are used to infer node embeddings. GraphSAGE employs the random walk algorithm (Perozzi et al., 2014) to generate context nodes for each target node and then adapts the negative sampling technique (Mikolov et al., 2013) to learn the model parameters.

GAT (Veličković et al., 2018) extends GCN in assigning importance weights to neighbors of a given node by exploring the standard attention technique (Bahdanau et al., 2015). GAT induces a new vector representation $\boldsymbol{h}_{\mathsf{v}}$ of node v from its neighbors as follows:

$$\boldsymbol{h}_{\mathsf{v}} = \mathsf{g}\left(\sum_{\mathsf{u} \in \mathcal{N}_{\mathsf{v}} \cup \{\mathsf{v}\}} \tau_{\mathsf{v},\mathsf{u}} \boldsymbol{W}\boldsymbol{x}_{\mathsf{u}}\right), \forall \mathsf{v} \in \mathcal{V} \tag{14}$$

where $\tau_{\mathsf{v},\mathsf{u}}$ is an importance weight which is computed as follows:

$$\tau_{\mathsf{v},\mathsf{u}} = \frac{\exp\left(\mathsf{LeakyReLU}\left(\boldsymbol{a}^{\mathsf{T}}\left[\boldsymbol{W}\boldsymbol{x}_{\mathsf{v}} \oplus \boldsymbol{W}\boldsymbol{x}_{\mathsf{u}}\right]\right)\right)}{\sum_{\mathsf{u}' \in \mathcal{N}_{\mathsf{v}}} \exp\left(\mathsf{LeakyReLU}\left(\boldsymbol{a}^{\mathsf{T}}\left[\boldsymbol{W}\boldsymbol{x}_{\mathsf{v}} \oplus \boldsymbol{W}\boldsymbol{x}_{\mathsf{u}'}\right]\right)\right)} \tag{15}$$

## B  EFFECTS OF HYPER-PARAMETERS

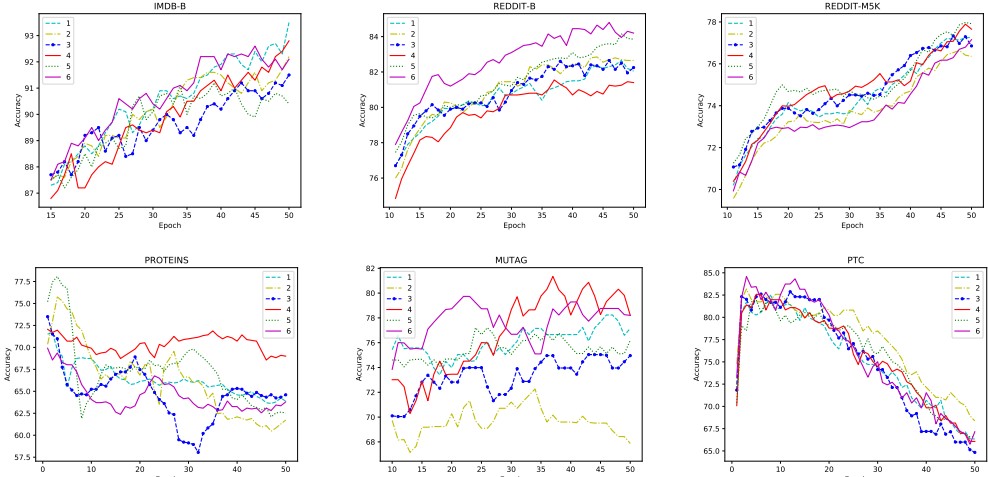

Figure 4: Effects of the number of steps ($T$) in the unsupervised fashion. Regarding each dataset, for all 10 folds, we vary the value of $T$ while using the same fixed values of other hyper-parameters.

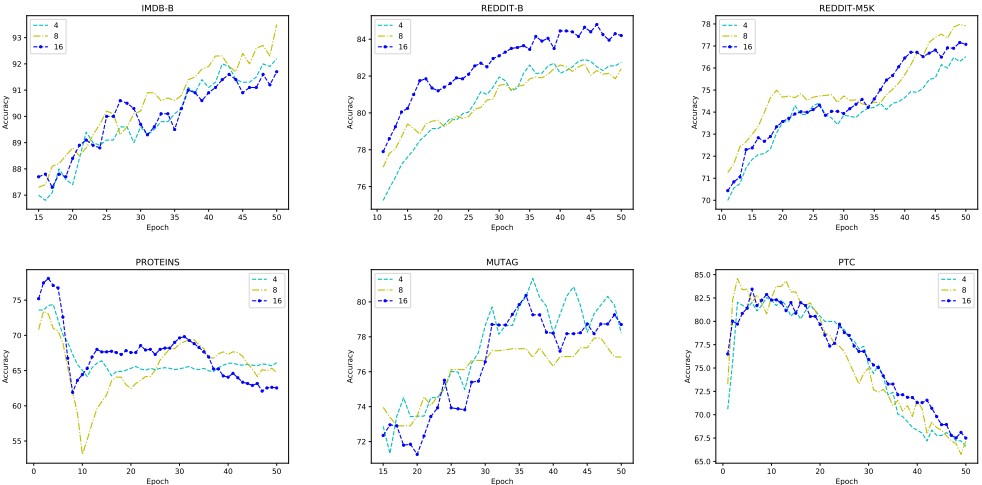

Figure 5: Effects of the number of neighbors ($N$) sampled for each node in the unsupervised fashion. Regarding each dataset, for all 10 folds, we vary the value of $N$ while using the same fixed values of other hyper-parameters.

