# OpenReview forum: "Unsupervised Universal Self-Attention Network for Graph Classification"
_ICLR.cc/2020/Conference — Reject_

### Official Review · AnonReviewer3 · 2019-10-23
**Official Blind Review #3**

**Rating:** 3

**Review:**

In this paper, the authors developed a graph embedding method called U2GAN based on self-attention mechanism. Similar to many existing graph neural network, U2GAN samples and aggregate neighboring features for each node in a graph. The aggregation function is similar to GAT, i.e., a query based attention layer. The difference is an incorporation of a transition function followed the attention layer. By minimizing Eq. 7, node embeddings can be inferred, which are summed up to obtain a graph embedding, for the downstream graph classification task.

There are several points that are unclear in the paper.
1.  The major argument of the advantage of using self-attention for neighborhood aggregation is to facilitate memorizing the dependencies between nodes and explore the graph structure similarities locally and globally. This argument, however, was not clearly discussed in the paper. First, it is not clear on why existing GNN methods, such as GCN, GraphSage, and GAT, cannot do so. Second, it is not clear on how does the proposed U2GAN achieve it. The current paper only provides some high-level descriptions. A more specific or theoretical discussion is desired.
2. Since the attention based aggregation is similar to GAT, a discussion on the difference is important.
3. Several model designs are not well justified. In Eq. 2, 3, the reason to employ Layernorm is missing. In Eq. 7, the intuition on how does the loss function help learn effective embeddings remains to be clarified. Also, it may be better to evaluate different pooling method to obtain graph embedding to justify the choice of sum in Eq. 1.
4. Since the proposed method aims to learn node embeddings in an unsupervised manner, it is better to see some descriptions on why graph classification was selected as the task in evaluation, instead of node classification.
5. In the experiments, some methods such as deepwalk, node2vec, graphsage and GAT are missing in comparison. In particular, due to the similarity between the proposed method GAT, it is interesting to evaluate GAT by replacing its supervised loss by Eq. 7 as a compared method. Moreover, in fig. 4, other visualizations of other methods can be compared to demonstrate the difference between the proposed method and others.


**Experience Assessment:**

I have published in this field for several years.

**Review Assessment: Checking Correctness Of Derivations And Theory:**

N/A

**Review Assessment: Checking Correctness Of Experiments:**

I carefully checked the experiments.

**Review Assessment: Thoroughness In Paper Reading:**

I read the paper thoroughly.

---

> ### Author Response · Authors · 2019-11-08
> **Responses to your comments**
>
> Thank you for your comments.
>
> For some clarifications:
>
> 1.	We find that the the propagating phase in the GNN approaches (e.g. GraphSAGE, GCN, GIN) is not advanced enough to be able to determine latent potential relationships among nodes. You can see the the propagating phase of GCN, GraphSAGE and GAT in Appendix A.
>
> 2.	GAT borrows the standard attention from [1] in using a single-layer feedforward neural network parametrized by a weight vector and then applying the non-linear function followed by the softmax function to compute importance weights of neighbors for a given node. Therefore, GAT is much different with the self-attention mechanism. You can see more details about GAT in Appendix A of our revision.
>
> [1] Neural machine translation by jointly learning to align and translate. ICLR 2015.
>
> 3.	 Xu et. al. [2] showed that the sum pooling performs better than the mean and max poolings. Finally, our current results with the sum pooling are very promising.
> [2] How Powerful are Graph Neural Networks? Xu et. al., ICLR 2019.
>
> 4.        To the best of our knowledge, our work is the first to show that a unsupervised model can noticeably outperform most of up-to-date supervised approaches by a large margin. This is important in both industry and academic applications in reality where expanding unsupervised GNN models is more suitable due to the limited availability of class labels.
>
> We plan to investigate the effectiveness of our model on other important tasks such as node classification and link prediction in the future work.
>
> 5.       As mentioned in our revision, our "supervised" model produces new state-of-the-art accuracies on DD, IMDBBINARY, IMDBMULTI, PROTEINS and PTC; and obtains competitive accuracies on remaining datasets.
>
> 6.	We included the results of GraphSAGE in our revision.  Existing works do not include DeepWalk or Node2Vec as baselines for the downstream task of graph classification.
>
> We do not have experimental results for GAT. But Shchur et. al. [3] showed that GCN outperforms GAT in different split settings.
>
> [3] Pitfalls of Graph Neural Network Evaluation. Shchur et. al. 2018.
>
> We did not include the embedding visualization of other baselines because those methods are very different from our unsupervised model.
>
> We are looking forward to your new updated comments on our revised submission.
>
> Thanks.

---

### Official Review · AnonReviewer2 · 2019-10-23
**Official Blind Review #2**

**Rating:** 1

**Review:**




The paper presents an new unsupervised model for graph classification. It borrows the idea from universal self-attention network and applies it to graph learning. It achieves surprisingly good results on benchmark datasets. Despite the good results, I do not think the technical quality is good enough to make it accepted. My concerns contain the following aspects:

1.	If we compare the proposed model with the graph attention networks (GAT), it just adds the recurrent transition and the layer normalizer, which are also from the universal self-attention. This makes the paper not novel enough.  Furthermore, adding these components are not so related to unsupervised learning, it does not add any value to the unsupervised learning strategy.
2.	The description of the unsupervised learning objective is not clear. From Algorithm 1, it seems $o_v$ is equal to $h_v^T$, I cannot understand the meaning of Eq. (7) at all.
3.	The results are too good to be true. Although we cannot judge it based on this belief, the authors have to convince the readers and explain how the huge performance gain is obtained (on some datasets U2GAN is even 27% higher than all of other methods).  I understand the experimental setting is transductive, but even that cannot explain everything. To justify the experiments, the authors need to do a lot of ablation study, such as comparing with supervised learning version of this model, while in the paper there is no ablation model to explain it.




**Experience Assessment:**

I have published in this field for several years.

**Review Assessment: Checking Correctness Of Derivations And Theory:**

I assessed the sensibility of the derivations and theory.

**Review Assessment: Checking Correctness Of Experiments:**

I carefully checked the experiments.

**Review Assessment: Thoroughness In Paper Reading:**

I read the paper at least twice and used my best judgement in assessing the paper.

---

> ### Author Response · Authors · 2019-11-08
> **Responses to your comments**
>
> Thank you for your comments.
>
> For some clarifications:
>
> 1.	GAT borrows the standard attention from [1] in using a single-layer feedforward neural network parametrized by a weight vector and then applying the non-linear function followed by the softmax function to compute importance weights of neighbors for a given node. Therefore, GAT is much different with the self-attention mechanism. You can see more details about GAT in Appendix A of our revision.
>
> [1] Neural machine translation by jointly learning to align and translate. ICLR 2015.
>
> 2.	Regarding the unsupervised fashion, we aim to maximize the similarity between h^T_v and the node embedding o_v of a given node v, and also to minimize the similarity between h^T_v and the embeddings of "negative nodes". In addition, h^T_v may depend on sampling the neighbors of node v. Hence, after training, we choose to use o_v as the final representation of node v.
>
> Regarding the supervised fashion where we do not need to learn the node embeddings "separately", we can use h^T_v as the final representation of node v (i.e., o_v = h^T_v) in order to produce the vector representation o_G.
>
> 3.	We shared our code and running command scripts to make sure: (i) you can verify that our implementation is correct, and (ii) you can reproduce our experimental results.
>
> As mentioned in our revision, our "supervised" model produces new state-of-the-art accuracies on DD, IMDBBINARY, IMDBMULTI, PROTEINS and PTC; and obtains competitive accuracies on remaining datasets.
>
> 4.	To the best of our knowledge, our work is the first to show that a unsupervised model can noticeably outperform most of up-to-date supervised approaches by a large margin. This is important in both industry and academic applications in reality where expanding unsupervised GNN models is more suitable due to the limited availability of class labels.
>
> We are looking forward to your new updated comments on our revised submission.
>
> Thanks.

---

### Official Review · AnonReviewer1 · 2019-10-24
**Official Blind Review #1**

**Rating:** 3

**Review:**

The submission proposes a graph neural network based on propagation with the attention mechanism. Then the output function uses the summation of node vectors to read out information about the graph.

 While the design is good, all components are all known techniques: the sampling procedure is like GraphSAGE; the propagation rule is similar to GAT, and the output function is wide uses in graph neural networks.

Critics:

The writing is not clear. At the top of page 4: quote: "... and produce an output sequence {h_vi^t}i=1^N+1". Do you keep only the vector h_v1 and throw away other vectors? Because you will also put v_2 at the center and compute its vector in a different self-attention computation. If this is the case, why not just say the output is h_v1? If this is not the case, then each node will have multiple vectors: one is computed when the node is at the center, and others are computed when the node is sampled as a neighbor.

Below Boris Knyazev has several comments, which are not well addressed by authors. There is a discussion about transductive learning and inductive learning. However, it seems the authors still don't know how to run inductive learning on the graph classification task (quote "... still do not have a standard inductive setting for the graph classification task where we only use a part of each graph during training."). Boris does not suggest to use part of each graph; instead, he suggests not using test graphs. I believe this is the standard practice in inductive learning (e.g. kernel methods).

Another comment from Boris about the case when T=1, and the response is "T=1 does not correspond to a single layer network". I don't understand the response either. When T=1, a node only gets information from its neighbors. It is similar to a one-layer GCN or GAT, in which a node also only gets information from its immediate neighbors.

I also don't understand why the author insists that the proposed model has a layer-based architecture. In my view, it is a graph neural network by the standard of propagation rule and output function.





**Experience Assessment:**

I have published in this field for several years.

**Review Assessment: Checking Correctness Of Derivations And Theory:**

I carefully checked the derivations and theory.

**Review Assessment: Checking Correctness Of Experiments:**

I assessed the sensibility of the experiments.

**Review Assessment: Thoroughness In Paper Reading:**

I read the paper thoroughly.

---

> ### Author Response · Authors · 2019-11-08
> **Responses to your comments**
>
> Thank you for your comments.
>
> For some clarifications:
>
> 1.	As described after Equation 6 in the manuscript, at the final step (the T-th step), each node v can have multiple vectors as you mentioned; and we consider to only use the vector (at the center) at the final step to infer the embedding of the node v.
>
> 2.	We’ve discussed with Borsi that the graph classification task and “Not using test graphs” do not means we are testing inductive setting like in the node classification task. We suggested Boris to point a paper which describes an inductive setting for the graph classification task as reference, but got no response for this.
>
> 3.	To have better comprehensive experiments, we have implemented our model in the supervised fashion, and posted the results on October 28. In the revised manuscript, we show that our supervised model produces new state-of-the-art accuracies on DD, IMDBBINARY, IMDBMULTI, PROTEINS and PTC; and obtains competitive accuracies on remaining datasets.
>
> We have also shared our code to make sure you can verify and reproduce our results.
>
> 4.	In the layer-based GNN architecture like GCN/GAT, we have to construct (k+1) layers (i.e., multiple layers stacked on top of each other) to reach to k-hops neighbors of a given node.
>
> In our proposed model, T is not equal to the number of GCN/GAT layers. Not only T=1, but also for any value of T, we update the vector representation of each node v by recursively propagating the representations of its neighbors. This is reason why we do not need to implement a layer-based GNN architecture like GCN/GAT.
>
> We are looking forward to your new updated comments on our revised submission.
>
> Thanks.

---

### Public Comment · ~Boris_Knyazev1 · 2019-10-04
**Impressive results, but I'm skeptical about the correctness of evaluation**

This is an interesting work that tries to unify the Transformers and Graph Networks and proposed a model called U2GAN. The quantitative results are impressive.

But it would nice if the authors could address the following concerns.

1. I'm not sure that the evaluation is correct. Please correct me if I'm wrong.
In Section 4.3, the authors say that "For each dataset, after obtaining the graph embeddings, we perform the same evaluation process from ... , which is using 10-fold cross-validation scheme to calculate the classification performance for a fair comparison."
It sounds like the authors first use the entire dataset (train+val splits) to learn node embeddings. Then you sum node embeddings for each graph to obtain graph embeddings. Then you split the dataset into the train and val splits and train an SVM. If that's the case, it's unfair compared to baselines which do not have access to test graphs during training.
The authors should first split data and then use only the training set to learn node embeddings.
Please clarify. It would be also nice to provide a link to the code to check correctness.

2. The self-attention mechanism looks very similar to attention over weights in Graph Attention Networks (GAT) [1]. In particular, Eq.(5,6) in this submission are basically the same as Eq.(3,4) in [1].
It's also very similar to Graph Transformer Network [2]. The difference and contribution compared to both papers should be discussed. Empirical comparison would be a bonus. One difference is unsupervised training in this submission VS supervised training in [1,2]. What are other differences?

3. Another concern is the results in Figure 3, where it's shown that even for T=1 the results are at least 92.5% for COLLAB and 91% for DD. But, as far as I understand, T=1 corresponds to a single layer graph network, in which each node has only access to its first-hop neighbors according to Section 3. How is it possible for a single layer network to perform so well?
Ablation studies should be performed to show the contribution of each component of the proposed model compared to some widely used baseline, such as GCN. How important is LayerNorm, residual connections?

4. Also, are weights W1 and W2 in Eq.(4) shared over all propagation steps t? What's the number of trainable parameters in your model? Is it comparable to baselines?
For baselines, the authors used results reported in previous works, but this can be problematic, because the network architecture and hyperparameters can be very different making the comparison unfair. A better way is to run baseline experiments with the same hyperparameters and an equivalent architecture (or number of trainable parameters).

5. It would be interesting to compare to "Deep Graph Infomax" [3], which is another way to train unsupervised. At least, the benefits of this submission compared to [3] should be discussed.

6. Figure 4 does not tell me much without a comparison to the baseline embeddings.

7. In Figures 3,5,6, results for N=4,8,16 for the bionformatics datasets and REDDIT should be very similar, because the graphs are very sparse. Why the results are different in some cases? I believe there is a lot of noise in those plots, so for a better comparison the plots should be computed for results averaged after multiple runs.

8. U2GAN sounds confusing, because of "GAN" which makes me think it is related to Generative Adversarial Networks, which is not.

Thanks.

[1] Petar Veličković, Guillem Cucurull, Arantxa Casanova, Adriana Romero, Pietro Liò, Yoshua Bengio, Graph Attention Networks, ICLR, 2018
[2] Yuan Li, Xiaodan Liang, Zhiting Hu, Yinbo Chen, Eric P. Xing, Graph Transformer Network, submitted to ICLR, 2019
[3] Petar Veličković, William Fedus, William L. Hamilton, Pietro Liò, Yoshua Bengio, R Devon Hjelm, Deep Graph Infomax, ICLR 2019

---

> ### Author Response · Authors · 2019-10-05
> **Address other concerns**
>
> Thank you for your other comments.
>
> #2. The attention mechanism used in GAT extends from the standard attention technique in [1] and it does not use the query, key, and value projection matrices to compute the attention weights.
>
> #3. It is to note that T=1 does not correspond to a single layer network as we do not construct a layer-based architecture like GCN/GAT.
>
> In our model, the representations of each node and its sampled neighbors are iteratively refined in each iteration. Therefore, each node can aggregate information from k-hops neighbors in subsequent iterations. Thus, our model work well even T=1. You can see our intuition in the last paragraph of page 3.
>
> #4. The weight matrices W1 and W2 in Eq. 4 are shared over all timesteps t.
>
> As I mentioned in the above comment, the evaluation protocol is fair enough as many previous works report the published results taken from the original papers.
>
> #2&5. Thank you for pointing out the Graph Transformer Network and the Deep Graph Infomax which we do not know yet.  The architecture of the Graph Transformer Network is to derive graph-to-graph mapping which is quite different from our model. We will read the Deep Graph Infomax paper and see whether we can discuss the differences.
>
> #7. The bionformatics datasets are sparse, but not for REDDIT (even when the average number of neighbors per node in REDDIT is around 2.3 as you see in Table 1) because there are central nodes which link to most of nodes in each graph, thus each graph in REDDIT is not sparse.
>
>
> [1] Neural Machine Translation by Jointly Learning to Align and Translate. ICLR 2015.
>
> We hope you are fine with our answers.
> Thanks.

---

> > ### Public Comment · ~Boris_Knyazev1 · 2019-10-08
> > **Thanks**
> >
> > 2. I think this should be discussed in the paper with more details.
> > 3. This still sounds very much like a single layer graph network . From the formulas I don't see a sequence anywhere except for "Coordinate embedding", which you only apply to MUTAG and PROTEINS.
> > 4. It still would be useful to know the number of parameters. It looks like V, Q and K matrices alone will have 3x1024x1024, i.e. 3 million, parameters, which seems to be quite a lot for datasets such as MUTAG with 188 graphs.
> >
> > Thanks.

---

> > > ### Author Response · Authors · 2019-10-09
> > > **K, Q and V are in R^{d x d}. d is the dimension of node feature vectors. not 1024.**
> > >
> > > Thank you very much for your comments.
> > >
> > > 2. We will discuss this in our paper if it is needful.
> > >
> > > 3. You can see the "Constructing U2GAN" paragraph in page 3, we uniformly sample a set of N neighbors for each v ∈ V, and then use node v and its N neighbors as an input sequence for the U2GAN learning process. We illustrate this in Figure 2.
> > > We emphasize again that we do not construct a layer-based architecture like general GNNs.
> > >
> > > 4. It is to note that K, Q and V are in R^{d x d}, in which d is the dimension of node feature vectors as shown in Table 1. For example, on MUTAG, K, Q and V are 7x7 matrices; and then W1, W2 are 1024x7 and 7x1024 matrices respectively.
> > >
> > > Thanks.

---

> ### Author Response · Authors · 2019-10-05
> **The evaluation is fair enough and correct**
>
> Thank you for your comments.
>
> We would say that our evaluation protocol is correct.
>
> For the graph classification task, it is to note that the unsupervised models do not use the graph labels during training, while the supervised models leverage on the graph labels to improve the accuracy performance.
>
> Therefore, many people (including you) usually see that the unsupervised models are outperformed by the supervised models, and then ignore a fact that the unsupervised models (including our model) can use the entire dataset ONLY w.r.t feature vectors and nodes, to learn graph embeddings. For more detail, you can check this fact in DGK [1], graph2vec [2], AWE [3] or in the code of another ICLR 2020 submission [4].
>
> We emphasize that the evaluation is fair enough; and other baselines and our model use the same 10-fold cross-validation scheme. In addition, we use the same data splits as used in [5,6] for our implementation. Our code is available only to Reviewers and ACs via a private comment.
>
> [1] Deep Graph Kernels. KDD 2015.
> [2] graph2vec: Learning Distributed Representations of Graphs. 2017.
> [3] Anonymous Walk Embeddings. ICML 2018.
> [4] HOW CAN WE GENERALISE LEARNING DISTRIBUTED REPRESENTATIONS OF GRAPHS? ICLR 2020 submission.
>
> [5] How Powerful are Graph Neural Networks? ICLR 2019.
> [6] An End-to-End Deep Learning Architecture for Graph Classification. AAAI 2018.
>
> To this end, we respectfully disagree with your suggestion that the unsupervised models (e.g., [1,2,3,4]) should only use the training set to learn graph embeddings. Your suggestion is unfair, not only for the unsupervised models in the graph classification talk, but also in other classification tasks such as node classification (e.g., DeepWalk, Node2Vec and LINE); and besides, you do not know a fact that some supervised models such as GCN/GAT also use feature vectors of nodes in the test set during training on Cora/Citeseer/Pubmed.
>
> Thanks.

---

> > ### Public Comment · ~Boris_Knyazev1 · 2019-10-08
> > **Thanks for a response**
> >
> > Leveraging information about test graphs, such as graph structure, node features, etc. (but of course not labels!) during training is often called transductive learning. This is what people often do in node classification tasks Cora/Citeseer/Pubmed. In graph classification and in some other node classification tasks, people usually solve the inductive learning task, where we assume that we don't know anything about test graphs.
> > Transductive and inductive learning methods should be compared separately, for example as in the "Deep Graph Infomax" paper in Table 2.
> > So, I think it would be nice to report the results with and without using test graph features. Otherwise, it's simply unclear where improvements are coming from.

---

> > > ### Author Response · Authors · 2019-10-09
> > > **Regarding the inductive setting on the node classification task**
> > >
> > > Again, thank you for your interests.
> > >
> > > Although the node classification task is out of domain in this topic, we are willing to discuss something.
> > >
> > > We would like to suggest you to read papers [1] and [2] to see a standard inductive setting on Cora/Citeseer/Pubmed rather than the "Deep Graph Infomax". Follow this standard inductive setting where "only a part of the input graph is used to train the node embedding model, and the trained model can be then used to infer embeddings for newly unseen nodes in the remaining part of the input graph", you can see in [2] that GCN gives a low performance on Cora/Citeseer/Pubmed. Last year, we found the similar low performance with GAT on Cora/Citeseer/Pubmed in this standard inductive setting. We will verify the paper you mentioned in this setting when we have time.
> > >
> > > [1] is the first paper working on the standard inductive setting on Cora/Citeseer/Pubmed, but this setting is less mentioned. And nowadays, many papers only rely on another inductive setting from the GraphSAGE paper on the PPI dataset.
> > >
> > > [1] Revisiting Semi-supervised Learning with Graph Embeddings. ICML 2016.
> > > [2] Learning Graph Representations with Embedding Propagation. NIPS 2017.
> > >
> > > Note that the experimental setting for the graph classification task is similar to the transductive setting for the node classification task. You can see that the overview in our result table is similar to Table 2 in both the GCN and GAT papers where reporting the unsupervised and supervised models together.
> > >
> > > To our best of knowledge, we still do not have a standard inductive setting for the graph classification task where we only use a part of each graph during training.
> > >
> > > Thanks.

---

> > > ### Author Response · Authors · 2019-10-09
> > > **Only focus on the graph classification task**
> > >
> > > Thank you for your comments.
> > >
> > > Your point means that we should compare our unsupervised model only with other unsupervised models such as DGK and AWE. We agree with this point as most of the unsupervised models do the same thing.
> > >
> > > However, most of the supervised models consider that it is fair to compare the results from the unsupervised and supervised models together, because of using the same 10-fold cross-validation scheme (Please look into the result tables in the original papers of the supervised models we refer in our paper). Therefore, this is reason why we follow to include the supervised models and nicely separate them as shown in our result table.
> > >
> > > Thanks.

---

### Public Comment · ~Nam_T._Pham1 · 2019-10-20
**Nice results. How about the training time?**

Hi, it's a nice work and easy to follow. However, I wonder the training time because of a very large number of parameters in Transformer-based architectures.

---

> ### Author Response · Authors · 2019-10-21
> **Training time is reasonable**
>
> Hi Pham, thank you very much for your comments.
>
> We note that node embeddings o are in R^d; weight matrices K, Q and V are in R^{d x d}; and weight matrices W1 and W2 are in R^{k x d} and R^{d x k} respectively. We fix k to 1024; and d is the dimension of node feature vectors as shown in Table 1 (i.e., d is small). Thus, the training time is reasonable. For example, we show the average of training time for each epoch in the below table.
>
> Dataset		|	Average time for each epoch (minutes)
> ------------------------------
> COLLAB		| 	2.66
> DD			|	1.40
> NCI109 		|	1.29
> NCI1 		|	1.27
> PROTEINS   	|	0.40
> IMDB-M 	|	0.20
> IMDB-B 		|	0.18
> PTC 		|	0.07
> MUTAG 		|	0.02
> ------------------------------
>
> Thanks

---

### Author Response · Authors · 2019-10-28
**Update our U2GAN in a "supervised" fashion**

To ensure fairness, we've followed several existing works in comparing supervised models with unsupervised models due to the use of the same 10-fold cross-validation scheme.
We observe that our unsupervised U2GAN significantly outperforms the supervised methods in most of benchmark cases.
However, in light of Boris Knyazev's comment that we should not directly compare together, we have decided to further implement our U2GAN in a "supervised" fashion where we additionally construct a single fully connected layer to predict the graph labels.
This attempts to further provide additional evidence for the use of our model in general setting (either supervised or unsupervised).
We will update the results of our "supervised" U2GAN when we finish training models for all 11 experimental datasets.

Thanks.

---

> ### Author Response · Authors · 2019-10-31
> **Update some initial results of our "supervised" U2GAN**
>
> We can see that our "supervised" U2GAN produces new state-of-the-art accuracies on DD, IMDBBINARY, IMDBMULTI, PROTEINS and PTC ; and obtains a competitive accuracies on MUTAG.
>
> Dataset			| 	Result (% accuracy)
> --------------------------------------------------------
> DD   			|	81.24 +- 1.84
> IMDBBINARY   	|	79.40 +- 4.35
> IMDBMULTI   	|	56.20 +- 3.35
> MUTAG   		| 	89.97 +- 3.65
> PROTEINS   		|	78.53 +- 4.07
> PTC   		   	|	79.36 +- 4.06
> -------------------------------------------------------
> We will update the final results when we finish training and tuning the model's hyper-parameters for all 11 datasets.

---

### Author Response · Authors · 2019-11-06
**Reviewers might have had influenced by Boris Knyazev's comment, completely ignored our contribution**

Dear the Area Chairs and the Program Chairs,

We share our code and running command scripts to make sure that our implementation is correct and reviewers can reproduce our experimental results.  We have followed several existing works in comparing supervised models with unsupervised models due to the use of the same 10-fold cross-validation scheme.

However, we feel that reviewers (especially AnonReviewer2 on "too good to be true") might have had influenced by Boris Knyazev's comment on the correctness of evaluation, and hence they might have had completely ignored our contribution where, to our best of knowledge, our work is the first to show that a unsupervised model can noticeably outperform most of up-to-date supervised approaches by a large margin.
As a consequence, we thought we might not be allowed to compare the supervised models with our unsupervised model together, hence we posted a new comment on October 28 about "Update our U2GAN in a "supervised" fashion".

We note that our most important contribution now is to suggest that future GNN works should pay more attention to the unsupervised fashion. This is important in both industry and academic applications in reality where expanding unsupervised GNN models is more suitable due to a limitation of available class labels.

We are updating our paper including the experimental results of our supervised fashion for our rebuttal. Recently, our "supervised" fashion produces new state-of-the-art accuracies on DD, IMDBBINARY, IMDBMULTI, PROTEINS and PTC ; and obtains a competitive accuracies on MUTAG.

We hope that the Area Chairs and the Program Chairs can guide reviewers to have constructive discussions with us in the rebuttal process. In addition, it is very grateful if the Area Chairs can find the 4-th reviewer who can verify our implementation and is firstly not biased by Boris Knyazev's comment.

Thank you very much.

---

> ### Author Response · Authors · 2019-11-08
> **Update our revised submission, experimental results in the supervised fashion**
>
> Dear the Reviewers, the Area Chairs and the Program Chairs,
>
> We updated our revised submission, in which we included our experimental results in the supervised fashion. As mentioned in our revision, our "supervised" model produces new state-of-the-art accuracies on DD, IMDBBINARY, IMDBMULTI, PROTEINS and PTC; and obtains competitive accuracies on remaining datasets.
> We are training and tuning our model hyper-parameters due to a limitation of time and computing resources, hence the current "supervised" results is not final.
>
> Our proposed model now works in both the supervised and unsupervised fashions, thus we changed our paper title to "A UNIVERSAL SELF-ATTENTION GRAPH NEURAL NETWORK".
>
> To the best of our knowledge, our work is the first to show that a unsupervised model can noticeably outperform up-to-date supervised approaches by a large margin.  Therefore, we suggest that future GNN works should pay more attention to the unsupervised fashion as well as not comparing supervised models with unsupervised models together.
> This is important in both industry and academic applications in reality where expanding unsupervised GNN models is more suitable due to the limited availability of class labels.
>
> We are looking forward to new comments from the the Reviewers as well as the Area Chairs on our revised submission.
>
> Thank you very much.

---

### Decision · Program_Chairs · 2019-12-19

**Decision:**

Reject

**Comment:**

All three reviewers are consistently negative on this paper. Thus a reject is recommended.